# Consequences of Industry 4.0 in Business and Economics

**Petra Maresova [1],\* , Ivan Soukal [1], Libuse Svobodova [1], Martina Hedvicakova [1] ,**
**Ehsan Javanmardi [2] , Ali Selamat [3] and Ondrej Krejcar [4]**

[1] Department of Economics, University of Hradec Králové, Rokitanskeho 62, 50003 Hradec Králové, Czech Republic; ivan.soukal@uhk.cz (I.S.); libuse.svobodova@uhk.cz (L.S.); martina.hedvicakova@uhk.cz (M.H.)

[2] School of Economics, Management and Social Sciences, Shiraz University, Shiraz 7194854358, Iran; s.ejavanmardi@rose.shirazu.ac.ir

[3] Malaysia Japan International Institute of Technology (MJIIT), Universiti Teknologi Malaysia Kuala Lumpur, Kuala Lumpur 54100, Malaysia; aselamat@utm.my

[4] Center of Basic and Applied Research, Faculty of Informatics and Management, University of Hradec Králové, Rokitanskeho 62, 500 03 Hradec Králové, Czech Republic; ondrej.krejcar@uhk.cz

\* Correspondence: petra.maresova@uhk.cz; Tel.: +420-737-928-745

**Abstract:** The introduction of information technology into all aspects of our lives has brought forth qualitative and quantitative changes on such a large scale that this process has come to be known as the Fourth Industrial Revolution, or Industry 4.0. The aim of this paper is to fill in the gaps and provide an overview of studies dealing with Industry 4.0 from the business and economic perspectives. A scoping review is performed regarding business, microeconomic and macroeconomic economic problems. Four investigators performed a literature search of the Web of Science, Scopus, and Science Direct. The selected period spanned from 2014 to 2018, and the following keywords were used for the search: Industry 4.0, economics, economic development, production economics, and financial sector. A total of 2275 results were returned. In all, 67 full papers were screened. Results obtained from the relevant studies were, furthermore, divided into the following categories: work and skills development; economy growth and macroeconomic aspect; sustainability; intelligent manufacturing; policy; and change in business processes. Findings show that the aspects of work and skills development, smart technology adoption, intelligent manufacturing, and digitalization are very well described. The government and its policies usually play the role of a needed supportive element. Usually studies lack a coherent view of the topic in question and solve partial questions.

**Keywords:** consequences of Industry 4.0; economics; business processes; economy growth

**JEL Classification:** M2; J4

## 1. Introduction

Originally initiated in Germany, "the Fourth Industrial Revolution", known as Industry 4.0, has attracted much attention in recent literature. Industry 4.0 is defined as "the integration of complex physical machinery and devices with networked sensors and software, used to predict, control and plan for better business and societal outcomes" (Industrial Internet Consortium 2017) or "a new level of value chain organization and management across the lifecycle of products" (Kagermann 2014) or "a collective term for technologies and concepts of value chain organization" (Hermann et al. 2016).

The global industrial landscape has changed radically in the last few years due to rapid technological developments and innovations in manufacturing processes (Pereira and Romero 2017).

The first three industrial revolutions have brought mechanization, electricity, and information technology (IT) to human manufacturing. The development and change that has have taken place in the industry recently entered a new phase in parallel with the developments in computer technology (Lasi et al. 2014).

It is too early to predict how global and local economies will deal with the consequences of Industry 4.0. No more than 7% of studies concerned with Industry 4.0 focus on the issue of sustainability. The concept of Industry 4.0 entails necessary changes in the operational processes of companies. However, the macro- and microeconomic points of view of Industry 4.0 remain a relatively little explored area. There exist studies focusing on innovation processes in companies, on the replacement of labor by capital, and the consequences in the increasing unemployment rates and globalization (Antony 2009; Saam 2008; Sala and Triyín 2018; Hedvicakova 2018). Overall companies, households and the public sector are facing a big challenge in the next generation with important economic consequences. There is, however, a lack of studies concerned with economic growth and process change.

The aim of this paper is, therefore, to fill in the gap and provide an overview of studies dealing with Industry 4.0 from the economic perspective defined by keywords such as economics, economic development, production economics, financial sector.

## 2. Theoretical Background

Currently, the industrial value creation is shaped by the development towards the fourth stage of industrialization, so-called Industry 4.0. Industry 4.0, referred to as the "Fourth Industrial Revolution", also known as "smart manufacturing", "industrial internet" or "integrated industry", is currently a much-discussed topic. It is assumed that Industry 4.0 supposedly has the potential to affect entire industries by transforming the way goods are designed, manufactured, delivered, and paid for (Stock and Seliger 2016; Hofmann and Rüsch 2017).

Industry 4.0 is the next step in a long process of development, a revolution based on the use of cyber-physical systems (Grieco et al. 2017). In fact, the consequence of developing the Internet of Things and Big Data is the conception of Industry 4.0 as a consequence of their continuous development (Witkowski 2017). Opportunities for further development and direction and visions related to Industry 4 are introduced by (Pfeiffer 2017). The main ideas of Industry 4.0 were originally published by Kagermann based on cyber-physical system-enabled manufacturing and service innovation during the Hannover Fair event in 2011 that resulted from an initiative regarding high-tech strategy for 2020 (Lee et al. 2014; European Commission 2018; Zhou et al. 2015) and informed the Industry Manifesto 4.0 released in 2013 by the Acatech Academy of National Science and Technology (Druckversion 2018).

As a first preliminary summary, we define Industry 4.0 as follows:

- Products and services are flexibly connected via the internet or other network applications, such as a block chain.
- Digital connectivity enables an automated and self-optimized production of goods and services, including deliveries, without human interventions (self-adapting production systems based on transparency and predictive power). The value networks are controlled in a decentralized manner, while system elements (like manufacturing facilities or transport vehicles) make autonomous decisions (Hofmann and Rüsch 2017).

Industry 4.0 is a broad term, and different authors interpret it in different contexts. The prevailing interpretation of the term, however, refers to new technologies, digitization, and robotization. Lu (2017) lists the following areas relevant to Industry 4.0: Internet of Things (IoT), cyber physical system (CPS), information and communications technology (ICT), enterprise architecture (EA), and enterprise integration (EI). Industry 4.0 will have a strong impact along whole value chains and provide a set of new opportunities regarding business models, production technology, creation of new jobs, work organization, and workflows (Pereira and Romero 2017; Erol et al. 2016; Schlechtendahl et al. 2015). Industry 4.0 will lead to fundamental changes in the economy, work environment, and skills development (Pereira and

Romero 2017). A major role is played also by concepts such as smart products, automation, new ways of communication, or creation of new business models.

Industry 4.0 provides new paradigms for the industrial management of small and medium enterprises (SMEs). Supported by a growing number of new technologies, this concept appears more flexible and less expensive than traditional enterprise information systems, such as ERP and MES. However, SMEs find themselves ill-equipped to meet these new opportunities regarding their production planning and control functions (Moeuf et al. 2018).

Research results indicate that strategic, operational, as well as environmental and social opportunities are positive drivers of Industry 4.0 implementation, whereas challenges regarding competitiveness, future viability, as well as organizational and production fit impede its progress. Moreover, it is shown that the perception of Industry 4.0-related opportunities and challenges, which is the first step toward the Industry 4.0 implementation, depends to a great extent on different company characteristics (Müller et al. 2018).

Smart products are integrated in the whole value chain as an active part of the operational systems, where companies are able to monitor their own production stages through data storage, request the required resources, and control the production processes autonomously (Pereira and Romero 2017). Smart products store the details of how they were manufactured and how they are intended to be used as they actively support the manufacturing process (de Man and Strandhagen 2017). These products are equipped with sensors, identifiable components, and processors which carry information and knowledge to convey functional guidance to customers and transmit the user feedback to the manufacturing system (Abramovici and Stark 2013). Industry 4.0 increases cost- and time-efficiency and improves product quality, which is associated with enabling technologies, methods, and tools (Albers et al. 2016).

Business models are greatly influenced by Industry 4.0, since this new manufacturing paradigm entails a new way of communication along supply chains (Glova et al. 2014). In the field of business, Industry 4.0 implies that a complete communication network will exist between various companies, factories, suppliers, logistics, resources, customers, etc. Each section optimizes its configuration in real-time depending on the demands and status of associated sections in the network, which generates maximum profit for all cooperatives with limited sharing resources (Kagermann et al. 2018). The problem of business models is solved in context of how a company's positioning helps to understand how to derive benefits from Industry 4.0. In particular, the positioning as user and/or provider of Industry 4.0 has a large impact on SME business models. The study suggests managers explore further forms of business model innovation, and create customer-driven, rather than product-oriented innovations (Müller et al. 2018). Another study deals with the fact that the industrial Internet of Things poses several implications on manufacturers in terms of economic, ecological, and social aspects referring to the triple bottom line of sustainable value creation (Kiel et al. 2017).

Customers are a key factor in every business model, and Industry 4.0 brings a set of advantages for them, improving communication along the value chain and enhancing the customer's experience (Zhong et al. 2017). It allows customers to order any function of products, with any number of functions or products, even if there is only one of a kind. In addition, customers could change their order and ideas at any time during the production, even at the last minute with no additional charge (Schlechtendahl et al. 2015). Skills development, which will lead to demographic and social changes, is one of the most important key factors for a successful adoption and implementation of the Industry 4.0 framework (Pereira and Romero 2017).

Industry 4.0 will lead to an increased automation of tasks, which means that workers should be prepared to perform new tasks. The same applies to engineering education, which holds a large potential to train professionals of the future and make them aware of new technological trends and opportunities. Finally, managers should also adapt their management strategy to the new market requirements (Erol et al. 2016).

It is expected that Industry 4.0 can have further consequences on management and future jobs, allowing the creation of new business models, which will have a large effect on industry and markets, ultimately affecting the whole product lifecycle, providing a new way of producing goods and doing business, allowing an improvement of processes and increasing the company's competitiveness (Pereira and Romero 2017).

The implications of Industry 4.0 for business model components have been identified as below, which makes it possible to determine different ways to transform outdated models. Firstly, an improvement of the traditional business model with an incremental innovation of both value creation and value delivery has been defined. Secondly, a diversification of the actual business model through the reconfiguration of value networked ecosystems has been described as a radical innovation. Finally, a new business model typology based on the smartization of products and services has been proposed (Ibarra et al. 2018). In the Industry 4.0 world, which is characterized by digitalizing and automating, sustainable business models exist but have not become mainstream (de Man and Strandhagen 2017). Also, Industry 4.0 is used for three, mutually interconnected factors:

1.  Digitization and integration of any simple technical–economical relation in complex technical–economical complex networks;
2.  Digitization of product and service offers;
3.  New market models.

All these human activities are interconnected with a lot of communication systems at the moment. The most promising technologies will be the Internet of Things (IoT), Internet of Services (IoS), and Internet of People (IoP) (Sala and Trivín 2018).

## 3. Methods

This scoping review is performed to provide an overview and summary of up-to-date studies focusing on Industry 4.0 in business and economics.

### 3.1. Search Strategy and Eligibility Criteria

During June 2018, four investigators performed a literature search of the Web of Science, Scopus, and Science Direct. The selected period spanned from 2014 to Q1 2018, and the following keywords were used for the search: Industry 4.0, economics, economic development, production economics, and financial sector.

A total of 122 studies were identified on the Web of Science. After excluding ineligible papers, a total of 68 studies were chosen for further analysis. The Scopus database yielded 261 results. In both cases, the focus was on reviews and original papers, and conference papers found in the results were included in Table 1. The search in the Science Direct database focused on the presence of keywords in the "Title, abstract, or keywords." A total of 2275 results were returned. The most fruitful year was 2015, while the following years saw a decline in the number of papers. Most of the results were conference papers, which was apparent when the results were ordered by relevance, and these are included in Table 2.

**Table 1.** Numbers of papers in all databases as a result of the query Industry 4.0 plus the keyword in each row.

| Keywords for Searching | 2014 | 2015 | 2016 | 2017 | 2018 | Total |
|---|---|---|---|---|---|---|
| Economics | 57 | 141 | 124 | 121 | 86 | 529 |
| Economic development | 138 | 525 | 441 | 371 | 219 | 1694 |
| Production economics | 16 | 38 | 38 | 51 | 38 | 181 |
| Financial sector | 23 | 104 | 53 | 55 | 19 | 254 |
| **Total** | **234** | **808** | **656** | **598** | **362** | **2658** |

Following the elimination of duplicates, the title, abstract, and keywords of each paper were analyzed. The distribution of papers by year and keywords (or properties) is illustrated in Table 1.

From a total of 2658 papers that were yielded by the search and corresponded to the chosen criteria, papers including four or more of the required properties were selected and further analyzed. Some of the papers proved to be false positive, that is, while they contained the appropriate properties, their content was not relevant to the topic. After the initial elimination of unsuitable papers, a total of 292 papers remained for further processing. Following the final manual text screening, there were 67 full-length papers left for analysis.

### 3.2. Data Extraction and Study Quality Evaluation

The eligible papers were processed by four researchers, who worked independently. For each paper, the following data were identified: author(s), title of the paper, country of publication, and type of publication. In order to be eligible for analysis, the papers had to meet the following criteria:

- Published after 2013;
- Focusing on the macro- and microeconomic perspectives;
- Dealing with questions concerning government policy;
- Discussing global, regional, and investment consequences of Industry 4.0, investments in high-tech fields;
- Discussing innovative approaches for managing operational processes of companies;
- Written in the English language.

A publication was excluded if the following criteria applied:

- Focusing on a specification and description of particular technological solutions;
- Discussing specific solutions for a particular sector;
- Written in a language other than English.

Figure 1 graphically shows the process of publication search, selection, and analysis.

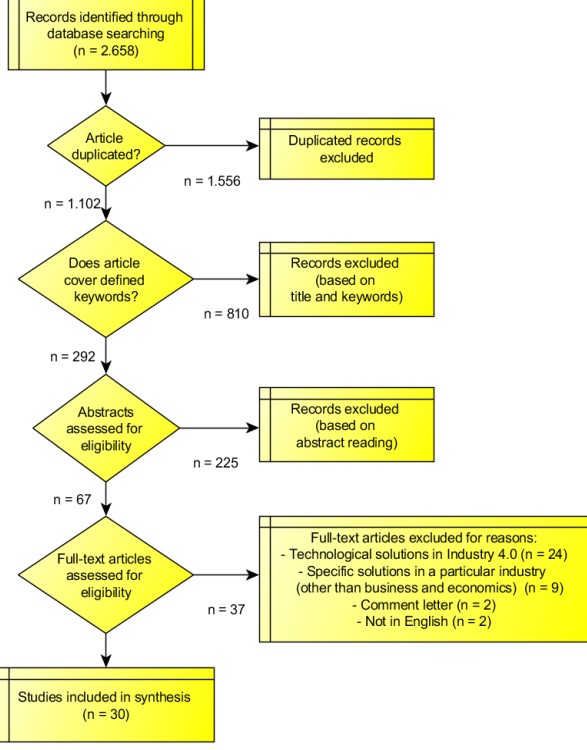

**Figure 1.** Publication search process.

**Table 2.** Aspects discussed in papers dealing with Industry 4.0 in the context of business and economy in research papers and proceedings papers.

| Research Papers | Country | Work Environment | Skills Development | Economy Growth and Macroeconomic Aspect | Sustainability/Environment | Digitalization/Smart Factory/Intelligent Manufacturing | Policy | Change in Business Systems/Processes |
|---|---|---|---|---|---|---|---|---|
| Agolla (2018) | x | | x | | | x | x | x |
| Akberdina et al. (2017) | | | | x | | | x | x |
| Gavrysh and Boiarynova (2017) | | | | | | | x | x |
| Hirsch-Kreinsen (2016) | | x | x | | | | x | x |
| Chovancova et al. (2018) | | x | | x | | x | | |
| Kireeva and Tsoi (2018) | | | | x | | | | |
| Petrillo et al. (2018) | x | | x | | | | | |
| Povolná and Švarcová (2017) | x | | | x | | x | x | |
| Prause and Atari (2017) | | | | | | | x | |
| Sung (2018) | x | | x | | | x | | x |

| Proceedings Papers | Country | Work Environment | Skills Development | Safety | Sustainability/Environment | Digitalization/Smart Factory/Intelligent Manufacturing | Policy | Change in Business Systems/Processes |
|---|---|---|---|---|---|---|---|---|
| Caricato and Grieco (2017) | | | | | | | | x |
| Elbestawi et al. (2018) | | x | | | | x | | |
| Erol et al. (2016) | | x | | | | x | | x |
| Gášová et al. (2017) | | x | | x | x | x | | |
| Chen (2017) | x | x | | x | | x | | x |
| Ibarra et al. (2018) | | | | | | x | | x |
| Issa et al. (2017) | | x | x | | | x | | x |
| Kuch and Westkämper (2017) | x | | | | x | | | |
| Meissner et al. (2017) | | | | | | | | x |
| Pagalday et al. (2018) | | x | x | | | x | x | x |
| Paravizo et al. (2018) | | | x | | x | | | |
| Pereira and Romero (2017) | | x | x | | | | | x |
| Prinz et al. (2017) | | x | x | | | x | x | x |
| Rauch et al. (2017) | | x | x | | | x | | x |
| Zhong et al. (2017) | x | | | | | x | | x |
| Reniers (2017) | | | x | x | | | | |
| Rita et al. (2017) | | x | x | | x | x | | x |
| Schumacher et al. (2016) | x | x | x | | x | x | | x |
| Sutikno and Suliswanto (2015) | x | | | | | x | x | |
| Tonelli et al. (2016) | x | | | | x | x | x | |

## 4. Results

The scoping review resulted in the selection of 30 papers, of which 20 were conference papers. All of these are listed in Table 2 and specified as to their focus according to the selected criteria, that is, the most discussed areas in the given papers.

With respect to education, not only skills of workers are a recurrent topic but also learning factories, which are connected with the concepts of digitalization and smart factory (Erol et al. 2016; Elbestawi et al. 2018; Issa et al. 2017; Prinz et al. 2017). Pereira and Romero (2017) state that skills development, which will lead to demographic and social changes, is one of the most important key factors for a successful adoption and implementation of the Industry 4.0 framework. He deals also with other impacts of Industry 4.0, such as products and services, business models and markets, economy, work environment, and skills development. Erol et al. (2016) is concerned with competences and a scenario-based learning factory approach. He describes the application of these concepts in the Industry 4.0 Learning Factory at TU Wien. Paravizo et al. (2018) explores gamification to support manufacturing education in Industry 4.0 as an enabler for innovation and sustainability. He discusses learning and education as well as game-based learning and gamification. Elbestawi et al. (2018) is concerned with existing learning factories that cover a variety of learning environments. Additionally, he describes the SEPT Learning Factory for Industry 4.0 Education and Applied Research. W Booth School of Engineering Practice and Technology (SEPT) is an educational unit at the Faculty of Engineering at McMaster University that has taken two significant steps in developing talents for a workforce that has Industry 4.0 foundational education and skills. Finally, educational trends related to Industry 4.0 are discussed. A similar topic is discussed by Prinz et al. (2017), who focuses on the implementation of a learning environment for Industry 4.0 assistance systems to improve overall equipment effectiveness.

A change in business systems and processes was one of the most frequently discussed topics. For instance, Rauch et al. (2017) examines critical factors for introducing lean product development in SMEs in Italy. He conducted qualitative research using a questionnaire which was filled in by 54 SMEs from various industries. Issues under discussion included: experience of SME with lean methods, success of lean projects in product development, application, efforts undertaken to introduce known lean methods and their potential and benefits, the influence of Industry 4.0 in the application of lean principles in product development, difficulties in the introduction of Industry 4.0 in product development, and the need for smart products (cyber-physical products). A maturity model for assessing Industry 4.0 readiness and the maturity of manufacturing enterprises is described by Schumacher et al. (2016). He examines nine dimensions, including strategy, leadership, customers, product, operations, culture, people, governance, and technology, and accompanies each dimension with exemplary maturity items. Ibarra et al. (2018) conducted a review focused on business model innovation through Industry 4.0. Tonelli et al. (2016) discusses novel methodology for manufacturing firms value modeling and mapping to improve operational performance in the Industry 4.0 era. Examined topics include above all "value creation," "process improvement," "maturity model," or "business model."

For a detailed description of the topic of each paper, particularly with respect to the main findings, only research articles published in academic journals were selected, which concentrate on a specific area in connection with the Industry 4.0 concept (Table 3).

**Table 3.** Summary of research paper studies.

| Title | Type of Study | Objective | Main Findings | Limits of the Study |
|---|---|---|---|---|
| Digitization of industrial work: development paths and prospects (Hirsch-Kreinsen 2016) | review | To provide an overview of preliminary results of the implementation of digital technologies in German industry. | The implementation of smart industry environments is paradoxical in nature in that it requires structural changes that present considerable challenges. | Lack of description. |

**Table 3.** *Cont.*

| Title | Type of Study | Objective | Main Findings | Limits of the Study |
|---|---|---|---|---|
| Fourth Industrial Revolution: Current Practices, Challenges, and Opportunities (Petrillo et al. 2018) | original | (1) To discuss the implications of Industry 4.0; (2) to illustrate with data the increased efficiency and productivity of companies using smart manufacturing. | Investment plans and best practice guidelines for implementing Industry 4.0 need to be developed on regional and national levels to facilitate the transition to smart systems. | Limited data access. |
| Human Capital in the Smart Manufacturing and Industry 4.0 Revolution (Agolla 2018) | review | To discuss the changing role of human capital in connection with Industry 4.0. | There will be an increasing need for qualified workers who are, besides their technical skills, creative and capable of working efficiently in smart environments. | Lack of description. |
| Changes in industrial structure of GDP and stock indices also with regard to the Industry 4.0 (Agolla 2018) | original | (1) To determine the position of individual industries in GDP and stock indices and (2) to predict changes in these positions in the Fourth Industrial Revolution. | (1) Demands on mechanical engineering are expected to increase to meet the need for new types of construction; and (2) engineers will be required to work more closely with the latest technologies, hence this area is expected to grow rapidly. | |
| Industry 4.0: A Korea perspective (Sung 2018) | original | (1) To provide a practical discussion of Industry 4.0 and (2) to propose policies for a transition to Industry 4.0 in Korea. | It is necessary to: (1) formulate government strategies; (2) establish an operational system to implement the policies; (3) create flexible actionable plans; and (4) set up infrastructure to manage the transition. | (1) The proposed policy implications are not backed by empirical evidence; (2) the Fourth Industrial Revolution and Industry 4.0 are used interchangeably, despite the differences between the two; (3) the findings are derived from literature reviews and government reports and cannot be yet supported by reliable statistics. |
| Mechanisms for Forming IT-clusters as "Growth Poles" in Regions of Kazakhstan on the Way to "Industry 4.0" (Kireeva and Tsoi 2018) | original | To provide theoretical and practical suggestions for creating IT-clusters in Kazakhstan as part of transition to Industry 4.0. | IT-clusters should be set up in several stages: (1) concentrating resources; (2) building an IT ecosystem; and (3) developing and maturing. | Lack of description. |
| On sustainable production networks for Industry 4.0 (Prause and Atari 2017) | original | To examine the relationships of sustainability and networking, structural conditions, and organizational development. | (1) The major obstacle to a reliable analysis that would allow drawing of general conclusions is the lack of data; (2) networked environments do not communicate as effectively to improve the performance of the structural unit or the organization. | Data availability. |
| The Driving Factors, Risks and Barriers of the Industry 4.0 Concept (Macurová et al. 2017) | original | To propose a tentative model of mixed types of stakeholder engagement approaches. | The most prominent feature of the current economic cycle with respect to resource markets is: (1) the marginalization of regional space in favor of the central space; and (2) integration of the periphery into central economic structures. | Resource markets in southern Russia are limited by fragmentation into multiple competing structures and small-scale commodity systems. |
| The Macroeconomic Context of Investments in the Field of Machine Tools in the Czech Republic (Povolná and Švarcová 2017) | original | To analyze production, export and import, and machine investments in the Czech Republic from the macroeconomic perspective with respect to Industry 4.0. | (1) Production and export are related to GDP fluctuations; while (2) investments in machine tools are independent of GDP. | Focuses on only one branch. |
| The methodological approach to monitoring of the economic and functional state of innovation-oriented machinery engineering enterprises at the modern technological modes (Gavrysh and Boiarynova 2017) | original | To develop a methodology for monitoring and evaluating the progress and success of innovative machine-engineering projects. | The proposed methodology involves the following stages: (1) developing the structure of indices to measure performance; (2) building an appropriate composition of machine-engineering innovation projects; (3) determining the dynamics of economic indices; (4) eliminating non-systemic influences; and (5) determining the desired value range of indices. | A small sample for verification. |

## 5. Discussion

What follows from the economic theories is that the implementation of new technologies and the substitution of labor by capital is a process taking place in all industries in order to reduce costs, increase productivity, and facilitate the provision of individual customer solutions. Most of the studies are concerned with manufacturing industries, although the Industry 4.0 initiative is relevant in all sectors. The focus was on identifying studies whose key topic is a discussion of the business

and economic implications of Industry 4.0. Such implications include: work environment, skills development, economy growth and macroeconomic aspect, sustainability and environment, policy, change in business processes, digitalization, smart factory, and intelligent manufacturing.

Our research found that the aspects of the work environment and skills development are very well described and monitored. Nevertheless, this does not mean that these issues are dealt with or even solved. Many studies stressed the increasing need for intelligent factory operators and the education process required for their training. Although it was not always the main contribution of the study, it was usually at least one of the most important prerequisites for a successful introduction of smart manufacturing, a new business process implementation, or other Industry 4.0 adoption. Above all, Petrillo et al. (2018) and then Agolla (2018), Hirsch-Kreinsen (2016) and Sung (2018) comment on the unsatisfactory situation concerning the lack of adequate education, which should start at high school and be realized through school–work alternation. In the current state of things, young workers are neither prepared for nor aware of the upcoming trend that they will most probably live and work in. Therefore, to tackle this issue, Petrillo et al. (2018) stress the need to create "systems behind the factory of the future" and promote internships.

Another problematic issue was related to workers who are already employed. In general, there is expected to be a potentially volatile situation when repetitive or routine-job workers will face a challenge to retain their jobs. Petrillo et al. (2018) sees the solution in continuous training; however, as Sung (2018) reminds us, retraining or even a new educational system does not solve the problem for older workers. Therefore, we assess this area of research as important, repeatedly stressed, but without a clear methodology as to how to resolve the issue. Many papers include suggestions for specific tools but do not include a framework; or, on the other hand, they provide general observations concerning the stages for dealing with the problem but do not elaborate these in detail. It appears that there is much space for further research, nevertheless, due to the very dynamic development of the area, it might not be completely clear what to include in the new education systems, except general IT and multidisciplinary skills development. It is highly probable that today's students will work in an industrial or service branch that does not exist yet or did not exist when they started attending school.

A great deal of discussion is focused also on the problem of smart technology adoption, intelligent manufacturing, and digitalization of industrial processes. One of the main recognized benefits is an ability to adapt faster to the rapidly changing environment. Hirsch-Kreinsen (2016) currently sees most promising adopters among mechanical engineering or logistic technology-intensive, strong mid-scale firms with the necessary qualified personnel and capabilities. This is because large-scale producers have already progressed very far in highly automated production technologies and organization, particularly in automotive and electro-technical sectors. Such companies might be, accordingly Hirsch-Kreinsen (2016), also cautious in smart manufacturing implementation because of certain skepticism concerning the efficiency promised by the smart systems. Also, the decentralized, automated self-organization nature of smart systems is far from current manufacturing and process standardization from which large automotive and electro-technical companies have been gaining a lot of profit so far. However, the process of adoption does not only facilitate production flexibility enhancement, there are also direct economic incentives in costs and efficiency that could eventually persuade even cautious adopters. As Petrillo et al. (2018) show, both these positive economic effects are present for European companies that have implemented smart manufacturing systems. Regardless of a company size, the studies described and stressed the creation of smart production networks with cyber-physical systems. The main advantage seen, e.g., by Prause and Atari (2017) is in achieving flexible and open value chains in the manufacturing of complex mass customization products in small series, which is not possible with current production ways and organizational structure. This is also closely related to another category: change in business systems and processes. In this category, one of the main general benefits of Industry 4.0 is seen in overall better planning and controlling (Petrillo et al. 2018).

The government and its policies are usually in the role of a needed supportive element or a required framework that should enable, enhance, and promote the Industry 4.0 adoption process in

numerous ways. In particular, for example, Sung (2018) or Petrillo et al. (2018), mention the national or regional policies to be set up:

- Education plans that pay much more attention to manufacturing topics and put more stress on foreign languages;
- Investment plans that encourage also middle-sized companies to adopt Industry 4.0;
- Plans for education focused on computer science and continuous education for the aforementioned intelligent factory operators;
- Policies to deal with the social structure problem due to low birthrate, income instability, work–family imbalance, etc.

Regarding possible future consequences in the area of economics or manufacturing, Industry 4.0 may contribute significantly to the continuing trend where a structure of industrial indices no longer corresponds to the GDP structure, which suggests that the stock market does not mirror the economy anymore (Chovancova et al. 2018). According to (Kireeva and Tsoi 2018), Industry 4.0 is a factor of growth by either the adoption of manufacturing technologies by existing companies or the formation of new ones in the IT industry. The findings of Beier et al. (2017) suggest that the technical transformation of industry digitalization is likely to be accompanied by social transformations. These social transformations will be reflected in the labor market, where the change of market needs will drive changes in educational systems, which is an oft-quoted aspect of Industry 4.0. Calculations made so far on the disappearance and creation of jobs vary with respect to the methodology used (Hedvicakova 2018). Osborne and Strokosch (2013) examine how susceptible jobs are to computerization. According to their estimates, about 47 % of total US employment is at risk. According to the analysis of Arntz et al. (2016), the ratio of vulnerable and newly opened jobs is 7:6 for the Federal Republic of Germany. As follows from a study prepared by SME-dominated mechanical and systems engineering industry organization in Germany and Europe VDMA (Der VDMA Verband Deutscher Maschinen- und Anlagenbau—VDMA (Der VDMA—VDMA 2018)), three main variants in the development of professional competences are likely:

- The growing gap scenario, in which the gap between poorly qualified and highly qualified laborers will continue to widen significantly;
- The general upgrade scenario, in which the demand for higher qualifications will be rising, and therefore everyone will be required to obtain advanced qualifications.
- The central link scenario, which stresses the need for higher qualifications and related highly specialized skills in qualified labor (Kagermann 2014).

As has been already stated, whether and to what extent some qualifications will become more or less valuable in future—and what kind of new qualifications will emerge—depends, among other things, on how fast and to what extent individual companies will implement automation and interconnect their procedures and processes in production, services, and sales. The relevant key competences of the future will be those pertaining to IT, software, application programs, and automated systems. These competences will involve not only basic know-how and ability to use digital devices, applications, Web 2.0, and any electronic tools, but also user-oriented skills will be required (CAD: Computer Aided Design, CRM: Customer Relationship Management, ERP: Enterprise Resource Planning) (National Institute for Education 2018).

Alongside specific professional qualifications and IT competences, more general skills and competences will also be increasingly important: communication skills, social skills, organizational skills, team work, project work, but also intercultural awareness and language skills. Last but not least, workers of the future will be expected to participate in life-long education to advance their skills and remain open to innovations (National Institute for Education 2018). The approach of society toward the concept of digital education should focus on new interdisciplinary study programs to cater for Industry 4.0 by combining the fields of mechanical engineering, electronic engineering,

and information technology and aiming to develop digital skills in the whole society, in connection with the digitalization of government administration and services for the employed, the unemployed, and those threatened by unemployment (Kuhnová 2017).

## 6. Conclusions

The current phenomenon of Industry 4.0 with respect to research studies focusing on business and economy culminated around the year 2015. The studies described the impacts of Industry 4.0 on the labor market, education, changes in operational processes, or economic growth. However, many studies lack a coherent view of the topic in question. The authors typically focus on one aspect of business and economy implications and continue to examine it in depth. The papers are usually based on the Industry 4.0 initiative but omit related initiatives, such as Work 4.0, Management 4.0, Marketing 4.0, and others. The interconnections and relationships of all relevant stakeholders, that is, private and state companies, the state, trade unions and employer unions, are often ignored. However, it is only by taking into account these very interconnections that individual countries can prepare for the social and economic impacts involved in the current trend of digitization and automation. In the future, an increasing interconnection of industry, science, research, and innovative new technologies can be expected, which needs to be approached in a complex way if the transition to Industry 4.0 is to succeed.

**Author Contributions:** Conceptualization and Methodology: P.M.; Data and Sources Searching: I.S., L.S., M.H., P.M.; Writing-Original Draft Preparation: I.S., L.S., M.H.; Writing-Review & Editing: M.H., E.J.; Visualization: O.K.; Supervision: A.S.

**Funding:** This research received no external funding.

**Acknowledgments:** The paper is supported by the internal project SPEV (2018) at the Faculty of Informatics and Management of the University of Hradec Kralove, Czech Republic. In addition, the authors thank Jan Hruska for her help with the research.

**Conflicts of Interest:** The authors declare no conflict of interest.

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
