# Peer review of "Consequences of Industry 4.0 in Business and Economics"

_economies, doi:10.3390/economies6030046_

Reviewer 1 Report

Unfortunately, the paper has several severe shortcomings:

Several statements are made regarding Industry 4.0 that are questionable (for example “revolving around sustainability”). Much more important, most of the statements made are not referenced. This applies for most of the introduction, foremost the second paragraph.

The term “Industry 4.0” is not defined properly. Central authors are not cited (see below), and the explanation where the term is coming from (German governmental initiative – Kagermann, 2013) is missing completely, among further aspects.

The main and vital flaw is the literature review: If the literature review was conducted in June 2018 (an earlier date could at least explain why some of the references from 2018 are missing), the literature review misses several vital authors in the field. On the basis of the described selection criteria, it is not understandable why, for example, the following authors are missing:

Beier, G., Niehoff, S., Ziems, T., & Xue, B. (2017). Sustainability aspects of a digitalized industry – A comparative study from China and Germany. International Journal of Precision Engineering and Manufacturing-Green Technology, 4(2), 227-234.

Lasi, H., Fettke, P., Kemper, H. G., Feld, T., & Hoffmann, M. (2014). Industry 4.0. Business & Information Systems Engineering, 6(4), 239-242.

Moeuf, A., Pellerin, R., Lamouri, S., Tamayo-Giraldo, S., & Barbaray, R. (2018). The industrial management of SMEs in the era of Industry 4.0. International Journal of Production Research, 56(3), 1118-1136.

Müller, J. M., Kiel, D., & Voigt, K. I. (2018). What Drives the Implementation of Industry 4.0? The Role of Opportunities and Challenges in the Context of Sustainability. Sustainability, 10(1), 247.

Stock, T., & Seliger, G. (2016). Opportunities of sustainable manufacturing in industry 4.0. Procedia Cirp, 40, 536-541.

At least 20 further central and high-quality sources are missing here. Consequently, the literature review seems to be conducted inappropriately, leading to incomplete results.

Based on the incomplete list of literature, the discussion and conclusion derive findings that cannot stand anymore, e.g., that only seven per cent of literature concerns sustainability aspects, among further.

Author Response

Comments and Suggestions for Authors

Unfortunately, the paper has several severe shortcomings:

Several statements are made regarding Industry 4.0 that are questionable (for example “revolving around sustainability”). Much more important, most of the statements made are not referenced. This applies for most of the introduction, foremost the second paragraph.

Authors: We agree that this part might have been rather misguiding. Therefore, we removed the part concerning sustainability in the introduction. We also added new references including the suggested ones.

The term “Industry 4.0” is not defined properly. Central authors are not cited (see below), and the explanation where the term is coming from (German governmental initiative – Kagermann, 2013) is missing completely, among further aspects.

Authors: we believe that now the "Industry 4.0" is defined properly and we added more key authors who contributed to the explanation of the term.

The main and vital flaw is the literature review: If the literature review was conducted in June 2018 (an earlier date could at least explain why some of the references from 2018 are missing), the literature review misses several vital authors in the field. On the basis of the described selection criteria, it is not understandable why, for example, the following authors are missing:

Beier, G., Niehoff, S., Ziems, T., & Xue, B. (2017). Sustainability aspects of a digitalized industry – A comparative study from China and Germany. International Journal of Precision Engineering and Manufacturing-Green Technology, 4(2), 227-234.

Lasi, H., Fettke, P., Kemper, H. G., Feld, T., & Hoffmann, M. (2014). Industry 4.0. Business & Information Systems Engineering, 6(4), 239-242.

Moeuf, A., Pellerin, R., Lamouri, S., Tamayo-Giraldo, S., & Barbaray, R. (2018). The industrial management of SMEs in the era of Industry 4.0. International Journal of Production Research, 56(3), 1118-1136.

Müller, J. M., Kiel, D., & Voigt, K. I. (2018). What Drives the Implementation of Industry 4.0? The Role of Opportunities and Challenges in the Context of Sustainability. Sustainability, 10(1), 247.

Stock, T., & Seliger, G. (2016). Opportunities of sustainable manufacturing in industry 4.0. Procedia Cirp, 40, 536-541.

At least 20 further central and high-quality sources are missing here. Consequently, the literature review seems to be conducted inappropriately, leading to incomplete results.

Based on the incomplete list of literature, the discussion and conclusion derive findings that cannot stand anymore, e.g., that only seven per cent of literature concerns sustainability aspects, among further. 

Authors:  

1.      We described in a greater detail our methodology, to be more specific our research timeframe ends at the first quartile 2018. Therefore, articles that were indexed in the databases after that date are unfortunately not included in our research.

2.      We understand and agree that Kagermann and several other authors are missing. This issue came from the scope and to be more specific from the keywords search itself.  Our goal was to find combine both „economics“ keyword and „industry  4.0“.  Therefore some, otherwise well-known contributions, were not included. Their keywords were mostly industry 4.0 + manufacturing; industry 4.0 + sustainability; industry 4.0 + factory; industry 4.0 + business etc. Nevertheless, we include now the most recognized authors in the paper. 

3.      In addition, we commented suggested studies also in the discussion.  

Reviewer 2 Report

Dear authors, the paper is interresting, please check references to be sure, that you are citing all publication which you use and there is no publication in references, which is not cited in the text.

The paper is just the assessment of the current state and there is nothing new/no special advice.

I suggest to extend this paper about suggestions what should happened in the area of education, what should happened in the area of economics or manufacturing etc.

This suggestions in the part of discussion/conclusion would be very helpful and improve this paper.

Author Response

Authors: we double-checked the references and corrected a few flaws (we used software Zotero now). We agree that our paper is an overview paper that does not seek to find solutions to each individual issue. Nevertheless, we extended the discussion by several points focused on what should happened in the mentioned area.

Reviewer 3 Report

The review of sources in Industry 4.0 is well done. Positive aspect of the contribution is that the analysis is not focused only on WoS and Scopus articles. The selection process of articles (methodology) is described on adequate level.

Conclusions are correct presented and discussion as well.

Author Response

 Dear Sir/Madam,

I am writing to you on behalf of the team that wrote an article titled Consequences of Industry 4.0 in business and economics. I  would like to thank for your comments and support.  

Yours sincerely,                                          

 Petra Maresova

Round  2

Reviewer 1 Report

Dear authors,

Thank you for your revision. While the first two issues (Introduction and Industry 4.0 definition) have been solved, the literature review still needs improvement:

1. After reading the reviewer response, it becomes clearer what you are aiming at. However, one does not really grasp that from reading the paper. A better explanation should be included why the keywords "Industry 4.0" and "economics" have been combined and why you focus on those and not others.

2. Several references, that should at least be in the discussion or introduction, are still missing. After a short literature search in Scopus, the top three findings for Industry 4.0 and economics (sorted by relevance, limited to business/economics as subject area) are:

Müller, J. M., Buliga, O., & Voigt, K. I. (2018). Fortune favors the prepared: How SMEs approach business model innovations in Industry 4.0. Technological Forecasting and Social Change, 132, 2-17.

Tortorella, G. L., & Fettermann, D. (2018). Implementation of Industry 4.0 and lean production in Brazilian manufacturing companies. International Journal of Production Research, 56(8), 2975-2987.

Sung, T. K. (2018). Industry 4.0: A Korea perspective. Technological Forecasting and Social Change, 132, 40-45.

Maybe, the second one really doesn`t fit, but number one and three do contribute to your research context and should therefore be included, e.g., in the discussion or introduction. 

Likewise, if you change "economics" to "economic", the top three references papers are:

Pfeiffer, S. (2017). The vision of “Industrie 4.0” in the making—a case of future told, tamed, and traded. NanoEthics, 11(1), 107-121.

Trantopoulos, K., von Krogh, G., Wallin, M. W., & Woerter, M. (2017). External knowledge and information technology: Implications for process innovation performance. MIS quarterly, 41(1), 287-300.

Kiel, D., Müller, J. M., Arnold, C., & Voigt, K. I. (2017). Sustainable industrial value creation: Benefits and challenges of industry 4.0. International Journal of Innovation Management, 21(08), 1740015.

Again, the second one doesn`t really fit, but 1 and 3 add to the research context. I suggest to include those, too.

Author Response

Authors:  Industry 4.0 is a concept that covers changes in technology, industrial production, digitization, etc. All of these sub-questions points to an economic growth. Furthermore, in the special issue stated "Companies, households and the public sector are facing and a big challenge in the next generation. These are changes brought about by the introduction of information technology into production, services and all sectors of the economy." Since we believe that the economic impacts are crucial for the whole society (and is not solved in a sufficient depth), we decided to focus on the above-mentioned keywords. We clarified this aim by adding an additional short explanation to the paper as well.

2. Several references, that should at least be in the discussion or introduction, are still missing. After a short literature search in Scopus, the top three findings for Industry 4.0 and economics (sorted by relevance, limited to business/economics as subject area) are:

Müller, J. M., Buliga, O., & Voigt, K. I. (2018). Fortune favors the prepared: How SMEs approach business model innovations in Industry 4.0. Technological Forecasting and Social Change, 132, 2-17.

Tortorella, G. L., & Fettermann, D. (2018). Implementation of Industry 4.0 and lean production in Brazilian manufacturing companies. International Journal of Production Research, 56(8), 2975-2987.

Sung, T. K. (2018). Industry 4.0: A Korea perspective. Technological Forecasting and Social Change, 132, 40-45.

Maybe, the second one really doesn`t fit, but number one and three do contribute to your research context and should therefore be included, e.g., in the discussion or introduction. 

Authors: Sung, T. K. (2018) – this source was present in the references in the previous version (please see ref. no. 39)

Müller, J. M., Buliga, O., & Voigt, K. I. (2018)  - unfortunately, this article was published after our review in July 2018 and so it is beyond our research time-frame. As we mentioned before, we made our methodology more clearly by stating that our search included articles indexed in the first quartile of 2018 and before. Nevertheless, we added now this paper into the “Theoretical background” since the contribution is valid and recent.

Likewise, if you change "economics" to "economic", the top three references papers are:

Pfeiffer, S. (2017). The vision of “Industrie 4.0” in the making—a case of future told, tamed, and traded. NanoEthics, 11(1), 107-121.

Trantopoulos, K., von Krogh, G., Wallin, M. W., & Woerter, M. (2017). External knowledge and information technology: Implications for process innovation performance. MIS quarterly, 41(1), 287-300.

Kiel, D., Müller, J. M., Arnold, C., & Voigt, K. I. (2017). Sustainable industrial value creation: Benefits and challenges of industry 4.0. International Journal of Innovation Management, 21(08), 1740015.

Again, the second one doesn`t really fit, but 1 and 3 add to the research context. I suggest to include those, too.

Authors:  we agree and we added to the “Theoretical background” comment and suggested sources Pfeiffer, S. (2017) and Kiel, D., Müller, J. M., Arnold, C., & Voigt, K. I. (2017). 

Round  3

Reviewer 1 Report

Dear Authors,

Thank you for your revision. I hope that the comments and review rounds were valuable for a) better explaining the research aim for readers and b) better position and discuss the paper in the existing literature.